# Salicylic and Methyl Salicylic Acid Affect Quality and Phenolic Profile of Apple Fruits Three Weeks before the Harvest

**DOI:** 10.3390/plants10091807

**Published:** 2021-08-30

**Authors:** Sasa Gacnik, Robert Veberič, Metka Hudina, Silvija Marinovic, Heidi Halbwirth, Maja Mikulič-Petkovšek

**Affiliations:** 1Department of Agronomy, Biotechnical Faculty, University of Ljubljana, SI-1000 Ljubljana, Slovenia; robert.veberic@bf.uni-lj.si (R.V.); metka.hudina@bf.uni-lj.si (M.H.); maja.mikulic-petkovsek@bf.uni.lj.si (M.M.-P.); 2Institute for Chemical, Environmental and Bioscience Engineering, Technische Universität Wien, A-1060 Vienna, Austria; silvija.miosic@tuwien.ac.at (S.M.); heidrun.halbwirth@tuwien.ac.at (H.H.)

**Keywords:** phenolics, bioactive compounds, phenylalanine ammonium lyase, chalcone synthase, chalcone isomerase, flavanone-3β-hydroxylase

## Abstract

Effects of spraying over apple trees (*Malus domestica*; ‘Topaz’) with methyl salicylic acid (MeSA) and SA during fruit maturation were investigated for quality parameters (weight, firmness, hue angle, red blush, yield) and phenolic profile of the peel and pulp (HPLC–mass spectrometry). These treatment effects were also investigated for activities of the phenylpropanoid pathway enzymes phenylalanine ammonia-lyase, chalcone synthase and isomerase (combined), and flavanone-3β-hydroxylase. The MeSA and SA treatments resulted in poor fruit peel coloration, with higher hue angles and 20% and 10% lower red blush, respectively. Anthocyanin levels were also significantly lower (56%) for MeSA treatment. MeSA stimulated activities of phenylalanine ammonia-lyase and chalcone synthase/isomerase, which resulted in higher levels of flavanols (to 34%), flavonols (to 33%), and hydroxycinnamic acids (to 29%), versus control. Therefore, while these salicylate treatments improve levels of some beneficial polyphenols, they also have negative effects on the external quality characteristics of the fruit.

## 1. Introduction

Salicylic acid (SA; also known as *ortho*-hydroxy benzoic acid) is an endogenous growth regulator and a signal molecule that is important for the induction of resistance to biotic and abiotic stress. In plants, it is present as the free phenolic acid and as conjugated forms that are generated by glycosylation, methylation, or hydroxylation of the benzoic ring [1]. Methyl SA (MeSA) is an inactive precursor of SA, and it has a key role in long-distance signaling from infected to non-infected tissue through the phloem [2].

Previous studies have reported the benefits of pre-harvest and post-harvest treatments with SA on various fruit quality characteristics, such as greater weight, fruit firmness and ascorbic acid content in peach fruit [3] and plum fruit [4], slower degradation of carotenoids in orange fruit pulp [5], and lower Croma index, higher total soluble solids, bioactive compounds, and antioxidant properties and better activity of some antioxidant enzymes in sweet cherry fruit [6,7].

Salicylic acid might have effects on the phenylpropanoid pathway through the induction of its key enzymes, such as phenylalanine ammonia-lyase (EC 4.3.1.5; PAL) and chalcone synthase (EC 3.2.1.74; CHS), which results in the accumulation of phenolic compounds [8,9]. Numerous studies have reported higher levels of total phenolic compounds and flavonoids in different SA-treated fruit, such as sweet cherries, [6] peaches [3], and apples [10]. SA treatments also increase the anthocyanin levels in apples [10], cornelian cherries [11], and raspberries [12], and the levels of individual flavonols (e.g., fisetin, morin) in ginger [9], and also of naringenin, one of the flavanones in ginger [9]. On the other hand, SA treatments can decrease carotenoid levels in apricots [13] and tomatoes [14], and flavonol (e.g., rutin) and flavone (e.g., apigenin) levels in ginger [9].

Due to public concerns about the negative impacts of synthetic chemicals on human health and the environment, there is ongoing research into new alternatives to the use of synthetic fungicides. One of the alternatives might be the application of SA, which has shown antifungal effects on some horticultural plants and fruit [15]. Yao and Shiping [16] reported SA efficacy against brown rot of sweet cherry fruit, which is caused by *Monilinia fructicola*. It is also known that SA has fungicidal effects on decay caused by *Botrytis cinerea* in strawberry fruit [15] and on anthracnose caused by *Colletotrichum gloeosporioides* in mango fruit [17].

The initial aim of the present study was to determine the effects of the exogenous applications over the last five weeks of maturation with MeSA and SA on apple (variety ‘Topaz’) fruit quality parameters (firmness, weight, hue angle, red blush, yield). We also investigated how long salicylates remained in the fruit peel after foliar applications to help determine the appropriate frequency of spraying. For more insight into the effects of these salicylates in the fruit peel and pulp, the levels of the major phenolic groups, the total phenolics analyzed, and the most important individual phenolics were also investigated. Finally, we also determined the effects on the activities of the key phenylpropanoid/ flavonoid-related enzymes: phenylalanine ammonia-lyase (PAL), chalcone synthase and isomerase combined (CHS/CHI), and flavanone-3β-hydroxylase (FHT). These data provide a further contribution to a better understanding of the mechanisms by which salicylates affect the quality of apple fruit.

## 2. Results and Discussion

### 2.1. Effects of MeSA and SA on External Quality Characteristics of the Apple Fruit

Several physiological and biochemical changes in fruit define the process of ripening. For apple fruit, these are expected to include decreased fruit firmness, increased total soluble solids and fruit weight, and changes in the color of the fruit peel [18,19]. Fruit weight, firmness, hue angle, red blush, and yield were determined here for these apples, as shown in Figure 1 and Figure 2.

The fruit firmness decreased for the shaded and sun-exposed sides of these fruits (Figure 1A), which is in agreement with Bizjak et al. [19]. The firmness of the fruit following the MeSA and SA treatments was significantly lower at day seven from each of these three treatments than for the control samples (*p* < 0.05) (Figure 1A, Spray#1-Spray#3). This finding is contrary to the reports by Mo et al. [20] for apples and Giménez et al. for cherries [7], who reported greater firmness in fruit treated with SA.

All of the timelines of these treatments showed increasing fruit weight over the five weeks from the first treatment (Spray#1) to the final analysis (Spray#3 plus 7 days). Figure 1B indicates that MeSA treatment resulted in significantly lower fruit weights at 7 days after each of these three treatments, compared to the controls (Spray#1–Spray#3). However, while the weights for the fruit following the SA treatment increased over these sampling times, the trend for lower weights than the control 7 days after spraying did not reach significance (*p* > 0.05). For the fruit following the MeSA treatment, the greatest weight difference compared to the control was seen 7 days after Spray#3 (13% lower than control; *p* < 0.05). In contrast, Martínez–Esplá et al. [4] and Giménez et al. [7] reported that treatments with MeSA and SA resulted in greater weights of plums and sweet cherries, respectively.

The treatments with MeSA and SA had an impact on the red color of the apple fruit, as shown in Figure 1C. The hue angle (h*) was used to define the red color of the apple fruit peel, and it was seen to decrease for the sun-exposed side during fruit maturation. 7 days after Spray#3, the control for the sun-exposed side of the fruit was 48% lower than for the first sampling time (i.e., prior to Spray#1; Figure 1C, T0). For the fruit following the SA treatment, the decrease in the hue angle for the sun-exposed side was similar to the control, at 45%, while for the fruit following the MeSA treatment, this decrease was significantly less than for the control, at 38% (*p* < 0.05) (Figure 1C). These data indicate that the MeSA treatments resulted in a significantly poorer red coloring of the fruit peel, with a similar trend seen for SA (*p* > 0.05). On the shaded side of the fruit, the treatment effects were lower, but again, generally showed significantly lower decreases in the hue angle compared to the control treatment (*p* < 0.05). Also, Tareen et al. [3] reported hue angles that were significantly higher for peach fruit treated with SA compared with control. The data in the present study indicate that the control apple fruit were significantly redder than the fruit following the MeSA treatment (with a similar trend following the SA treatment), which is one of the more desirable properties of this ‘Topaz’ cultivar. Similar data were obtained by Martínez–Esplá et al. [21] for plum fruit.

The lightness coefficient (L*) and Chroma (C*) parameters were also determined here, although no significant differences were seen between the treatments (data not shown). According to Shafiee et al. [22] and Supapvanich et al., [23], SA has no impact on the surface lightness of strawberry and wax apple fruit, respectively, while the hue angles of their SA-treated fruit were higher than the controls.

Treatments with MeSA and SA significantly reduced the increased red blush of the apple peel under the control treatment (*p* < 0.05) (Figure 1D). Thus, the control apples showed the highest red blush for all sampling times. At 7 days after Spray#1, the fruit with MeSA and SA treatments had 20% and 10% lower red blush, respectively, compared with the control, with this maintained through the other sampling times (Figure 1D, 7 days after Spray#2, Spray#3).

Previous studies have shown that salicylates can enhance strawberry and tomato fruit yields [24,25,26]. For the apple fruit here, there were no significant differences among these treatments in terms of the total yield per tree. However, there were significant differences among the treatments when the apples were classified into different quality classes according to the degree of red blush as follows: Class I, >70% red blush; Class II, 50% to 70% red blush; Class III, <50% red blush (Figure 2). The majority of the fruit following the MeSA and SA treatments were classified as Class II (Figure 2C,D 72%, 61%, respectively). This means that a large proportion of the total yield following the MeSA and SA treatments of the apple trees was in the second quality group (9 kg, 8 kg per tree, respectively) (Figure 2A). In comparison, the control apple trees had the highest proportion of fruit in the highest red blush group, as Class I (Figure 2B, 50%), for a yield of 6.7 kg Class I apples per tree (Figure 2A). The MeSA treatment here had the lowest yield for fruit in Class I (Figure 2C, 14%), at 1.9 kg per tree (Figure 2A), and the highest yield for fruit in the lowest quality class, Class III (Figure 2C, 14%), at 2.8 kg per tree (Figure 2A). Based on these data, the salicylate treatments reduced the proportion of the fruit in the highest quality group in terms of the red color of the fruit.

The total soluble solids and sugar and organic acid levels in the fruit pulp were also determined, but no significant differences were seen between these treatments (data not shown).

### 2.2. Effects of MeSA and SA on Accumulation of Phenolic Compounds in the Apple Peel

The phenylpropanoid pathway is one of the most important metabolic pathways in plants. Among the other metabolites, this pathway provides the different flavonoid classes, which are considered highly desirable compounds in the pharmaceutical, cosmetic, and nutraceutical industries [27]. The present study thus focused on the responses to the MeSa and SA treatments in terms of the levels of some specific phenolic groups in the apple fruit peel, using HPLC-MS analysis. The main representative groups of the phenolic compounds quantified in the apple fruit peel under these treatments for the different sampling times are given in Table 1. The data for the levels of the individual phenolic compounds of the most abundant phenolic groups (flavanols, flavonols, and hydroxycinnamic acids) at the sampling after Spray#1 are given in Appendix A.

For the initial chemical analysis of the apple fruit peel following the MeSA and SA treatments of the apple trees, Figure 3 shows that MeSA was better absorbed than SA, although it was then released more quickly, as seen particularly following the first sampling (6 h after Spray#1). Initially, here the MeSA level in the fruit peel was almost 6-fold higher than SA (Figure 3, 3.10 vs. 0.55 µ/g DW, respectively).

It has been reported that treatments with SA can increase the accumulation of flavonoids in different plants through triggering increased gene expression or increased activity of the gene products of various enzymes involved in the plant flavonoid pathways, such as CHS, CHI, flavanone-3-hydroxylase, and anthocyanidin synthase [28,29] Other studies have also reported effects of SA on flavonols. Ghasemzadeh et al. [9] reported a strong association of SA with increased levels of individual flavonols in ginger, including fisetin and morin.

In the present study, for the HPLC analysis of the apple fruit peel, 31 phenolic compounds were identified, which belonged to five specific groups: eleven flavanols, seven flavonols, six HCAs, three dihydrochalcones, and four anthocyanins (Table 1; Appendix A).

The flavanols (Table 1) represented the highest proportion of the phenolic compounds in the fruit peel (46–67% total phenolics analyzed). This is in agreement with Bizjak et al. [18], who reported the flavanol levels in the peel of ‘Braeburn’ apples as 50% to 65% of total phenolics analyzed. The flavanol levels in the fruit peel following Spray#1 here for MeSA and SA showed initial increases to 6 h post-spraying, with the maxima recorded at 24 h post-spraying (MeSA, 4.65 ± 0.50 mg/g DW; SA, 4.03 ± 0.12 mg/g DW), which then decreased to 48 h and 7 days after Spray#1 (Table 1). The spraying with MeSA had a significant effect on the accumulation of flavanols in the fruit peel compared to the control treatment, with a 34% increase over the control; there was also a trend to an increase for SA (16%), although this did not reach significance over the control. MeSA and SA also had similar effects on procyanidin levels in the fruit peel (Figure 4A), which were the most abundant of the flavanols total here (79–85%). However, the only significant difference between the treatments was at 24 h after Spray#1. Here, the procyanidin levels in the fruit peel for the MeSA treatment were 37% higher than in the control (*p* < 0.05), with an indication of 26% higher for the SA treatment (*p* > 0.05). Flacone Ferreyra et al. [30] reported that higher levels of phenolic compounds are associated with defense responses against pathogens, which are triggered by cell-signaling molecules. Sun et al. [31] showed that wild apple accessions resistant to blue mold had higher procyanidin levels than the commercial susceptible cultivar ‘Golden delicious’. This indicates that procyanidins might have one of the key roles in plant resistance against some pathogens. With higher procyanidin levels reached in the fruit peel here with the MeSA treatments (although only at 24 h post-spraying), the apples might show better resistance to plant pathogens. This aspect requires further detailed analysis.

The flavonols in the fruit peel represented the second-highest phenolic group (13–26% total phenolics analyzed). The flavonols analysis covered seven different types of quercetin glycosides (Appendix A), where the most abundant were quercetin-3-*O*-rhamnoside (55–73% total flavonols) and quercetin-3-*O*-galactoside (14–27% total flavonols). Similar to the flavanol levels, following Spray#1, the flavonol levels in the fruit peel increased from 6 h, with the maxima at 24 h, which then decreased (Table 1). Across the different treatments at the different sampling times, the only significant difference seen was for MeSA treatment 7 days after Spray#1, when the flavonol levels were 33% higher compared to both SA treatment and the control (*p* < 0.05). These changes in the flavonol levels also appear to be strongly correlated with the decreasing MeSA and SA levels in the fruit peel after the treatments (Figure 3).

The hydroxycinnamic acids in the fruit peel represented 2% to 3% of the total phenolics analyzed (6.90 ± 0.26 mg/g DW), which is similar to that reported by Mayr et al. [32]. The total HCA levels in the fruit peel were relatively constant over the 7 days following Spray#1. However, a small but significant increase was seen for the HCA levels of the fruit peel for the MeSA-treated apple trees by 24 h following spraying (Table 1). Here, the fruit peel contained 20% higher HCA levels compared to control (*p* < 0.05), as also for 7 days after Spray#1, as 29% higher (*p* < 0.05). The HCA levels between these treatments for the other sampling times did not differ significantly (*p* > 0.05). Chlorogenic acid was the major HCA in the fruit peel, at 57% to 76% of total HCA levels (Figure 4B; Appendix A). The changes in chlorogenic acid levels in the fruit peel for the control, MeSA, and SA treatments of the apple trees were similar to those of total HCAs (Figure 4B). The chlorogenic acid levels in the control fruit peel ranged from 107.2 ± 10.6 to 134.2 ± 1.7 g/kg DW, which tallies with the levels reported by Kalinowska et al. [33]. These control levels were not significantly affected by the SA treatment, while for MeSA, the chlorogenic acid levels were only significantly greater than the control at 24 h (38%; *p* < 0.05) and 7 days (29%; *p* < 0.05) after Spray#1 (Appendix A).

Anthocyanins are responsible for the red and blue colors in plants [32]. The anthocyanin profile (Table 1) of the fruit peel here mainly comprised the four cyanidin derivatives analyzed here: cyanidin-3-*O*-xyloside, cyanidin-3-*O*-arabinoside, cyanidin-7-*O*-arabinoside, and cyanidin-3-*O*-galactoside (data not shown). The changes in the color of the fruit peel following MeSA treatment of the apple trees, and the similar trends seen for SA, were supported by the measurements of the physical parameters of the hue angle (Figure 1C) and the red blush (Figure 1D), which appear to be connected with the changes in the anthocyanin levels across the sampling times (Table 1). MeSA had significant negative effects on the anthocyanin levels in the fruit peel at 24 h and 7 days after Spray#1, with the greater difference between the control and the MeSA treatment seen for the latter. At 7 days, the anthocyanin levels in the fruit peel following the MeSA treatments of the apple trees (0.39 ± 0.07 g/kg DW) were significantly lower than in the control fruit peel (0.90 ± 0.04 g/kg DW) by 57% (*p* < 0.05). This was paralleled by the SA treatment (0.67 ± 0.08 g/kg DW), with a smaller but not significant change in anthocyanin levels (26%; *p* > 0.05) compared to the control. Such lower anthocyanin levels in the fruit peel from the treated trees would indeed result in higher hue angles (Figure 1C) and lower red blush (Figure 1D) in comparison to the control treatment. Usually, studies report an increased anthocyanin content when treated with SA [11,12]. However, there may also be a drop in content in non-stressful conditions. This was shown by the results of a study by Bukhat et al. [34], where they found lower anthocyanin levels in SA-treated plants on non-saline soils, while the use of SA on stressed plants (saline soils) increased the anthocyanin content. They explained that stress factors increased the accumulation of ROS and that, for the detoxification of ROS radicals, SA would increase the production of anthocyanins as non-enzymatic antioxidants. When there is no need to degrade ROS, levels of stress-induced anthocyanins do not increase. The plants in our experiment were optimally grown, so stressful conditions should not have occurred. That in our case, the anthocyanins in the treated apples did not rise can be attributed to this phenomenon. Similar effects to those of MeSA and SA on the total anthocyanins were also reflected in all of the individual anthocyanin representatives here (data not shown), as represented by the cyanidin-3-*O*-galactoside levels shown in Figure 4C. This was the most abundant of the anthocyanins in the fruit peel (91–96% of total anthocyanins).

### 2.3. Enzymatic Activities

Investigation of the key enzymes of the phenylpropanoid pathway is needed to provide a better understanding of the defense mechanisms of plants against various biotic and abiotic stress factors. To better explain the increases in the phenolic compounds after spraying with these salicylates, the activities of selected enzymes of the phenylpropanoid pathway were determined as PAL, CHS/CHI combined, and FHT. Figure 5 shows these activities for the fruit peel samples following the control, MeSA, and SA treatments of the apple tree at the different sampling times.

Phenylalanine ammonia-lyase is one of the key enzymes in the phenylpropanoid pathway as it catalyzes the deamination of L-phenylalanine to trans-cinnamic acid. PAL is also a critical enzyme for plant responses to various stresses and exposure to phytohormones, such as SA [35,36]. During certain stress conditions, activation of PAL has been shown to increase the synthesis of compounds of the phenylpropanoid pathway [31]. The specific activity of PAL in the fruit peel following the control and SA treatments of the apple trees generally did not differ significantly, except at 6 h post-spraying, with similarly significantly lower activities for both compared to before spraying (Figure 5A; −92%, −89%, respectively). This correlates well with the low SA levels in the fruit peel through the sampling times (Figure 3), with the highest SA levels at 6 h post-spraying (0.5 µg/g DW) and the lowest at 7 days post-spraying (0.08 µg/g DW). In contrast, MeSA treatment of the apple trees increased the specific activity of PAL in the fruit peel at 6 h post-spraying by 15% compared to before spraying, thus representing a significant 14-fold increase in PAL activity over the control treatment at 6 h post-spraying (*p* < 0.05). A continued increase was seen to 24 h after MeSA treatment, where PAL activity in the fruit peel was 55% higher than before spraying, which was maintained to 48 h, and then lost by 7 days post-spraying. These data demonstrate a connection between the PAL activity and the MeSA levels in the fruit peel within the first 24 h, along with the HCA and flavonol levels at 24 h after MeSA treatment (Table 1, Figure 5A). Tang et al. [37] reported that PAL activity in poplar seedlings was also highest when treated with MeSA (as 1.5-fold the control).

Treatment of the apple trees with MeSA also increased the combined activities of CHS and CHI post-spraying, and more so over the control treatment, while SA treatment had little or no effect (Figure 5B). These enzymes catalyze the formation of flavanones, such as naringenin. This was shown previously by Ghasemzadeh et al. [9], who also reported that the CHS activity of ginger is concentration-dependent.

Positive effects of MeSA also emerged in FHT activity, which provides the precursors for flavonol formation [38]. FHT activity in the fruit peel following the control treatment of the apple trees was relatively unchanged across the sampling times, although a small but significant increase was seen at 6 h and 24 h (Figure 5C; *p* < 0.05). Interestingly, the increase in FHT activity for MeSA treatment over the control at 6 h and 24 h post-spraying was also seen for SA. There was also a connection between the FHT activity and the fruit peel content of these salicylates (Figure 3) within the first day of the treatments. MeSA and SA increased the FHT activities at 6 h post-spraying by 91% and 81%, respectively, compared to before spraying, which lasted to at least 24 h post-spraying.

## 3. Materials and Methods

### 3.1. Plant Material

This study was carried out on fruit of the apple (Malus domestica) variety ‘Topaz’ grafted onto M9 rootstocks in an experimental orchard of the Biotechnical Faculty in Ljubljana (latitude: 46° 02.906′ N; longitude: 14° 28.764′ E).

### 3.2. Treatments and Sampling

Fifteen randomly chosen apple trees that were similar in size, structure, and fruit-bearing received the experimental treatments of foliage and fruit spraying with a hand gasoline sprayer on three separate occasions in 2018: 22 August (Spray#1); 10 September (Spray#2); and 20 September (Spray#3). These treatments were with water (control), 2 mM MeSA, and 2 mM SA, 1 L/tree. For each of the treatment dates, 15 randomly selected fruit were collected from each of these trees before spraying and after each spraying (Spray#1-Spray#3), at 6 h, 24 h, 48 h, and 7 days. The fruit quality in terms of the external characteristics of fruit firmness, weight, color (hue angle, red blush), yield, and dry weight were initially determined. The chemical characteristics of the fruit were also determined as sugars and organic acids in the fruit pulp and phenolic compounds in the fruit peel. The pulp was sampled for the fruit collected at 7 days after each spraying (Spray#1–Spray#3), and the fruit peel for those collected for Spray#1 before spraying and at 6 h, 24 h, 48 h, and 7 days after spraying. For anthocyanin determination, the 7 days samples were also included for Spray#2 and Spray#3. Eight samples per treatment for each sampling time were frozen in liquid nitrogen and stored at –20 °C for analysis of the primary and secondary metabolites and at –80 °C for analysis of selected enzyme activities of the phenylpropanoid/flavonoid pathway.

### 3.3. Fruit Quality and Colour Development

For the external quality characteristics of the fruit, each was weighted to a precision 0.001 g. The fruit firmness was determined for both the shaded and sun-exposed sides of each fruit, using an electric pressure tester with a 10-mm tip, with these data expressed as kg cm^−2^.

To evaluate the red blush of the fruit surface, 15 randomly selected fruit per treatment were assessed visually from 0% (no red blush) to 100% (full red color) for each sampling time. Fruit color (L*, h*, C*) was determined on both the shaded and sun-exposed sides of the fruit using a portable colorimeter (CR-200b Chroma; Minolta, Osaka, Japan). The hue angle (h*) is expressed in degrees from 0° to 360°, where 0° represents red, 90° yellow, 180° green, and 270° blue. The L* represents the relative lightness of the colors and varies from 0, as black, to 100, as white. A digital refractometer (WM-7; Atago, Tokyo, Japan) was used for the soluble solids measurements (°Brix). The yields for each tree were also evaluated, and the fruit were classified into three quality classes: Class I, >70% red blush; Class II, 50% to 70% red blush; Class III, <50% red blush. After the classification, the fruit were counted and weighed.

### 3.4. Chemicals and Standards

For quantification of flavanols: procyanidin B1, catechin, and epicatechin (Fluka Chemie, Buchs, Sankt Gallen, Swiss). For hydroxycinnamic acids: p-coumaric acid, sinapic acid (Fluka Chemie, Buchs, Sankt Gallen, Swiss), chlorogenic acid, and 4-caffeoylquinic acid (Sigma-Aldrich, St. Louis, MO, USA). For flavonols: quercetin-3-rutinoside, quercetin-3-*O*-galactoside, quercetin-3-*O*-glucoside, quercetin-3-*O*-xyloside, kaempferol-3-*O*-glucoside (Fluka Chemie, Buchs, Sankt Gallen, Swiss), quercetin-3-*O*-arabinopyranoside and quercetin-3-*O*-arabinofuranoside (Apin Chem, Newbury, Berkshire, UK), isorhamnetin-3-*O*-glucoside (Extrasynthèse, Genay, France) and quercetin-3-*O*-rhamnoside (Sigma-Aldrich, St. Louis, MO, USA). For dihydrochalcones: phloretin and phloridzin (Fluka Chemie, Buchs, Sankt Gallen, Swiss). For anthocyanins, for quantification according to standard curves of cyanidin-3-*O*-galactoside (Sigma-Aldrich, St. Louis, MO, USA). For tissue extraction: methanol (Sigma-Aldrich, St. Louis, MO, USA) and formic acid (Fluka Chemie, Buchs, Sankt Gallen, Swiss). Double distilled water was used for all aqueous solutions. For HPLC mobile phases, acetonitrile (Sigma Aldrich, St. Louis, MO, USA), formic acid (Fluka Chemie, Buchs, Sankt Gallen, Swiss), and sulfuric acid (Sigma-Aldrich, St. Louis, MO, USA).

### 3.5. Extraction and HPLC-MS Analyses

For analysis of the phenolic compounds in the fruit peel, 2 g samples were extracted with 7 mL 80% methanol with 3% formic acid in a cooled (2° C) ultrasonic bath (Sonis 3, Iskra PIO, Šentjernej, Slovenia) for 45 min. The samples were then centrifuged at 3000× *g* for 5 min at 4 °C (5819 R; Eppendorf, Hamburg, Germany) and filtered into vials (0.20 µm; Chromafil AO-20/25 polyamide filters; Macherey-Nagel, Dueren, Germany).

Quantification of the individual phenolics was performed using HPLC (Thermo Scientific; San Jose, CA, USA), with the mobile phases of 97% acetonitrile + 0.1% formic acid and 3% acetonitrile + 0.1% formic acid, as described by Gacnik et al. [39]. Identification of the phenolic compounds was performed by mass spectrometry (LTQ XLTM Linear Ion Trap mass spectrometer; Thermo Scientific, Waltham, MA, USA) with electrospray ionization. The scans were from *m*/*z* 115 to 1500, operating in negative ion mode for the phenolics, except the anthocyanins, where positive ion mode was used. The diode array detector was set for three wavelengths: 280, 350, and 530 nm. For the analysis of the phenolic compounds, a C18 column was used (Gemini; 150 × 4.6 mm^2^, 3 μm; Phenomenex, Torrance, CA, USA), set at 25 °C. Quantification of the phenolics in the samples was achieved using the relevant calibration curves of external standards and the peak areas of the corresponding phenols. Anthocyanins were quantified according to standard curves of cyanidin-3-*O*-galactoside. Data are expressed in mg/g dry weight (DW) of fruit peel.

### 3.6. Sample Preparation for Enzyme Activity Assays

The activities of PAL (EC 4.3.1.5), CHS (EC 3.2.1.74), CHI (EC3.2.1.14), and FHT (EC 1.14.11.9) were analyzed for the peel from the fruit following the control and MeSA and SA treatments. The sample preparation for all of these enzymes was according to Halbwirth et al., [40] with minor changes.

Briefly, for each sample, ~0.4 g fruit peel frozen in liquid nitrogen was ground and homogenized in a ceramic mortar with 0.2 g quartz sand (Sigma-Aldrich, St. Louis, MI, USA), 0.2 g Polyclar AT, and 3 mL Dellus buffer, which contained 0.1 M HEPES (4-(2-hydroxyethyl)-1-piperazineethanesulfonic acid), 40 mM sucrose, 0.75 mM polyethylene glycol 20,000, 0.1 M sodium ascorbate, 1 mM dithioerythritol, and 0.025 mM CaCl2 (pH 7.3). Before use, the oxygen was removed from the buffer. Here, the mixture HEPES and sucrose were prepared to 150 mL with bi-distilled water and boiled for 20 min to a final volume of 100 mL. This was cooled in an ice bath to 30 °C under a nitrogen flow, and then the polyethylene glycol, sodium ascorbate, and dithioerythritol were added [41]. The preparation of the fruit peel was completed by centrifugation at 13,000× *g* for 10 min at 4 °C.

### 3.7. Crude Extracts Preparation

The crude extracts of the fruit samples for the enzyme assays were prepared according to the following protocol: after the centrifugation, 400 µL supernatant was passed through gel chromatography columns (Sephadex G25 medium). For analysis of PAL and CHS/CHI combined, these columns were eluted with 400 µL 0.1 M KPi with 0.4% ascorbate (pH 7.5). For analysis of FHT, the columns were eluted with 400 µL 0.1 M TRIS/HCl with 0.4% ascorbate (pH 7.5).

### 3.8. Enzyme Assays

The procedures for the PAL, CHS/CHI combined, and FHT assays followed those of Halbwirth et al. [40] with minor modifications. For the PAL activity assay, 40 µL crude extract was mixed with 55 µL 0.1 M KPi plus 0.4% ascorbate buffer (pH 8.5) and 5 µL [14C]-phenylalanine (0.027 nmol; 462,5 Bq). For the combined CHS/CHI activity assays, 40 µL crude extract was mixed with 50 µL 0.1 M KPi plus 0.4% ascorbate buffer (pH 7.5), 5 µL 1 nM p-CuCoA and 5 µL [14C]-malonyl-CoA (1.5 nmol; 1300 Bq). For the FHT activity assay, 40 µL crude extract was mixed with 50 µL 0.1 M Tris/HCl buffer plus 0.4% ascorbate (pH 7.5), 5 µL 2-oxoglutarate (1.46 mg m/L in H2O), 5 µL FeSO4 × 7H2O (0.56 mg/mL in H2O), and 5 µL [14C]-naringenin (0.036 nmol; 100 Bq). The assays were incubated for 30 min at 30 °C and then stopped with 200 µL ethyl acetate and 10 µL acetic acid (for PAL, CHS/CSI) or with 70 µL ethyl acetate and 10 µL acetic acid (for FHT). These samples were then centrifuged at 13,000× *g* for 3 min at 24 °C. For the PAL and CHS/CHI assays, 100 µL supernatant was transferred into scintillation vials and mixed with 4 mL scintillation cocktail (Rotiszint Eco Plus LSC, Carl Roth, Germany) and quantified using a scintillation counter (Wallac, Turku, Finland). For the FHT assay, 100 µL supernatant was transferred to thin-layer chromatography plates (1.05715.0001, Merck, Darmstadt, Germany). The plates were allowed to dry and then run overnight in a container with a chloroform/acetic acid/water solvent system (10:9:1; *v*/*v*/*v*). The conversion rates were then determined with a thin-layer chromatography linear analyzer (LB 284, Berthold, Bad Wildbad, Germany).

These enzyme activities are given as nanokatals (10–9 moles/s) per kg protein, with quantification according to Sandermann and Strominger, [42] using bovine serum albumin as the standard.

### 3.9. Statistical Analysis

The following statistical software were used to analyze the data: R-commander (R Formation for Statistical Computing, Auckland, New Zealand) and Statgraphics Plus 4.0 (Manugistics, Inc., Rockville, MD, USA). The individual effects of the MeSA and SA treatments on the fruit quality characteristics and on the secondary metabolites in the fruit peel underwent a one-way analysis of variance (ANOVA). The significant differences among the treatments were estimated using Duncan’s multiple range tests (*p* < 0.05). The specific enzyme activities were calculated, and the initial sampling (before spraying control) time for each enzyme was set to 100%. All of the other sampling times were calculated as specific enzymatic activity changes (%) relative to this control.

## 4. Conclusions

In conclusion, these data demonstrate some promising effects of the application of these salicylates to apple trees over the final few weeks of fruit maturation. These include an influence on the enzymatic activities in the fruit peel of some of the phenylpropanoid enzymes (i.e., PAL, CHS, CHI, FHT), and consequently changes in the accumulation of some of the important phenolic groups, such as flavanols, flavonols, hydroxycinnamic acids, and within these, some individual phenolic compounds, such as procyanidins, chlorogenic acid, and cyanidin-3-*O*-galactoside. Higher phenylpropanoids levels have been shown to contribute to improved resistance of plants to stressors and also to increase protection against plant diseases. However, we also show that these salicylates can have negative effects on the external quality characteristics of the fruit following these treatments. In particular, this included reduced red color (i.e., higher hue angle) and red blush of the fruit peel, as well as anthocyanin levels, following these treatments of the apple trees. Along with size, weight, and health status, the red coloration of apples is one of the important parameters for the classification of these fruit into different quality classes, which then determines their market value. Through the reduction of the red coloration of the fruit, these salicylates reduced the overall fruit yield per tree for the highest quality fruit, which consequently lowers the profit margin of the fruit producer. The effects of MeSA and SA are not promising since they decreased the number of fruit in Class I. Thus, they cannot be recommended for practical purposes, at least for this apple cultivar and at the applied dose. Other doses should be tested in future research in order to find beneficial effects in terms of increasing phenolic content without detrimental effects on fruit color. In order to obtain more precise recommendations on when to treat and with what concentration of salicylates, additional studies should also be carried out on other apple cultivars. Based on our results, it appears that the greatest effect on phenolic content occurred 24 h after treatment with MeSA.

## Figures and Tables

**Figure 1 plants-10-01807-f001:**
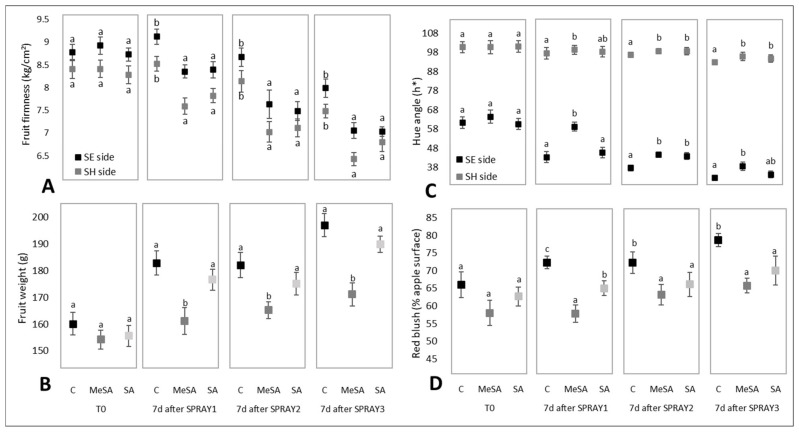
External quality characteristics of the apple fruit before any sprayings (T0) and 7 days after Spray#1 (SPRAY1), Spray#2 (SPRAY2), and Spray#3 (SPRAY3) for the control (**C**) and methyl salicylic acid (MeSA) and salicylic acid (SA) treatments. (**A**) Firmness on the shaded (SH) and sun-exposed (SE) sides of the fruit. (**B**) Fruit weight. (**C**) Peel hue angle on the shaded and sun-exposed sides of the fruit. (**D**) Peel red blush. Data are means ± standard error (*n* = 15). Different letters indicate significant differences among the treatments within each characteristic and each sampling time (*p* < 0.05; Duncan tests).

**Figure 2 plants-10-01807-f002:**
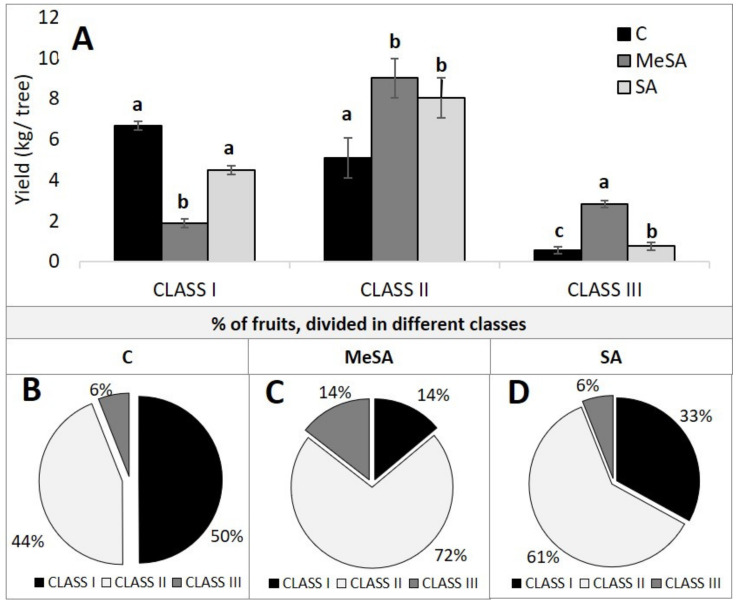
(**A**) Final yields of the apple fruit for the control (**C**), methyl salicylic acid (MeSA), and salicylic acid (SA) treatments, defined according to the three different quality classes. Data are means ± standard error (*n* = 15). Different letters indicate significant differences among the treatments within each class (*p* <0.05; Duncan tests). (**B**–**D**) Proportions of the apple fruit yields within each quality class for the control (**B**,**C**), methyl salicylic acid (MeSA) (**C**), and salicylic acid (SA) (**D**) treatments. Class I, >70% red blush; Class II, 50% to 70% red blush; Class III, <50% red blush.

**Figure 3 plants-10-01807-f003:**
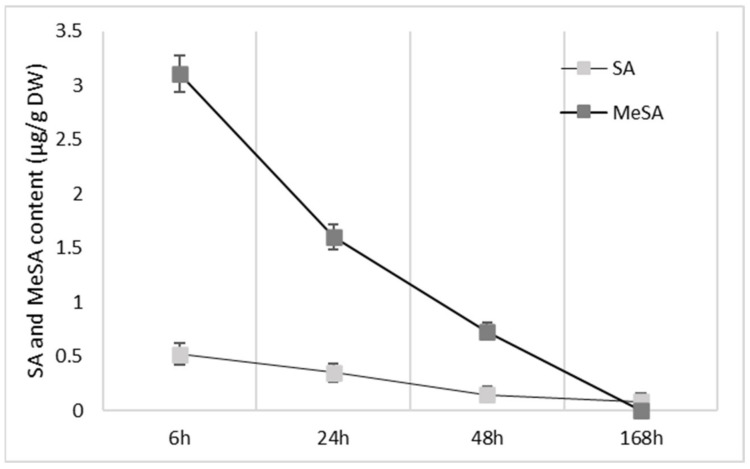
Content of methyl salicylic acid (MeSA) and salicylic acid (SA) in apple peel for the different sampling times after the first spraying (Spray#1). Data are means ± standard error (*n* = 10). DW, dry weight.

**Figure 4 plants-10-01807-f004:**
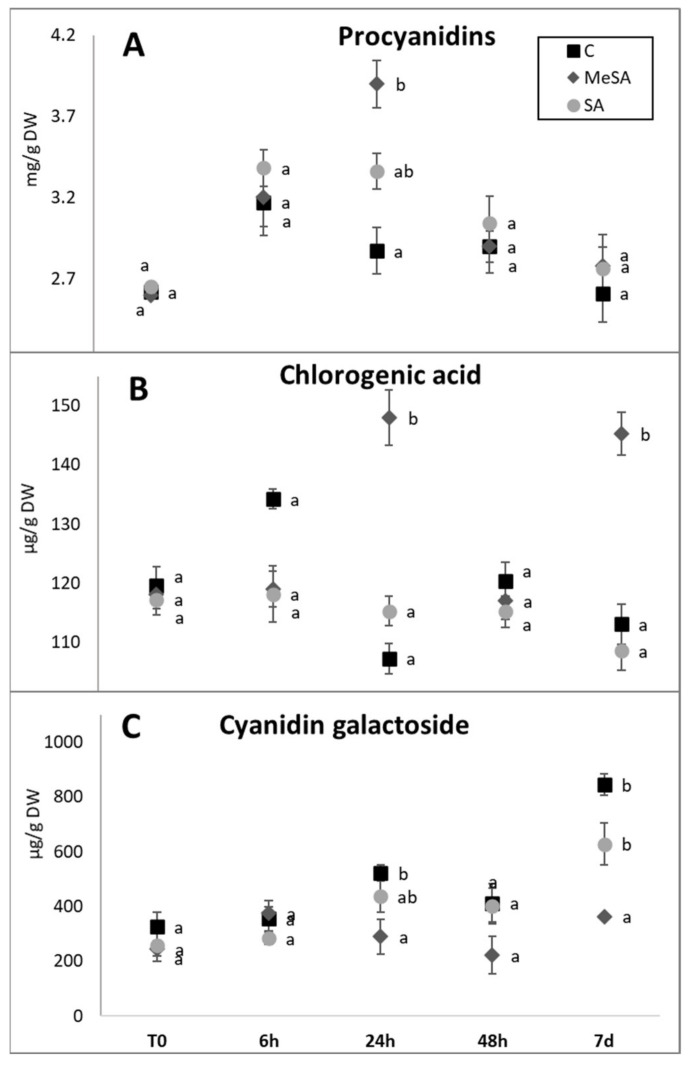
Procyanidins (**A**), chlorogenic acid (**B**), and cyanidin-3-*O*-galactoside (**C**) levels in apple fruit peel following treatments of apple trees, as control (**C**) and methyl salicylic acid (MeSA) and salicylic acid (SA) treatments at different sampling times after the first spraying (Spray#1). Data are means ± standard error (*n* = 10). Different letters indicate significant difference between treatments at each sampling time (*p* < 0.05; Duncan tests).

**Figure 5 plants-10-01807-f005:**
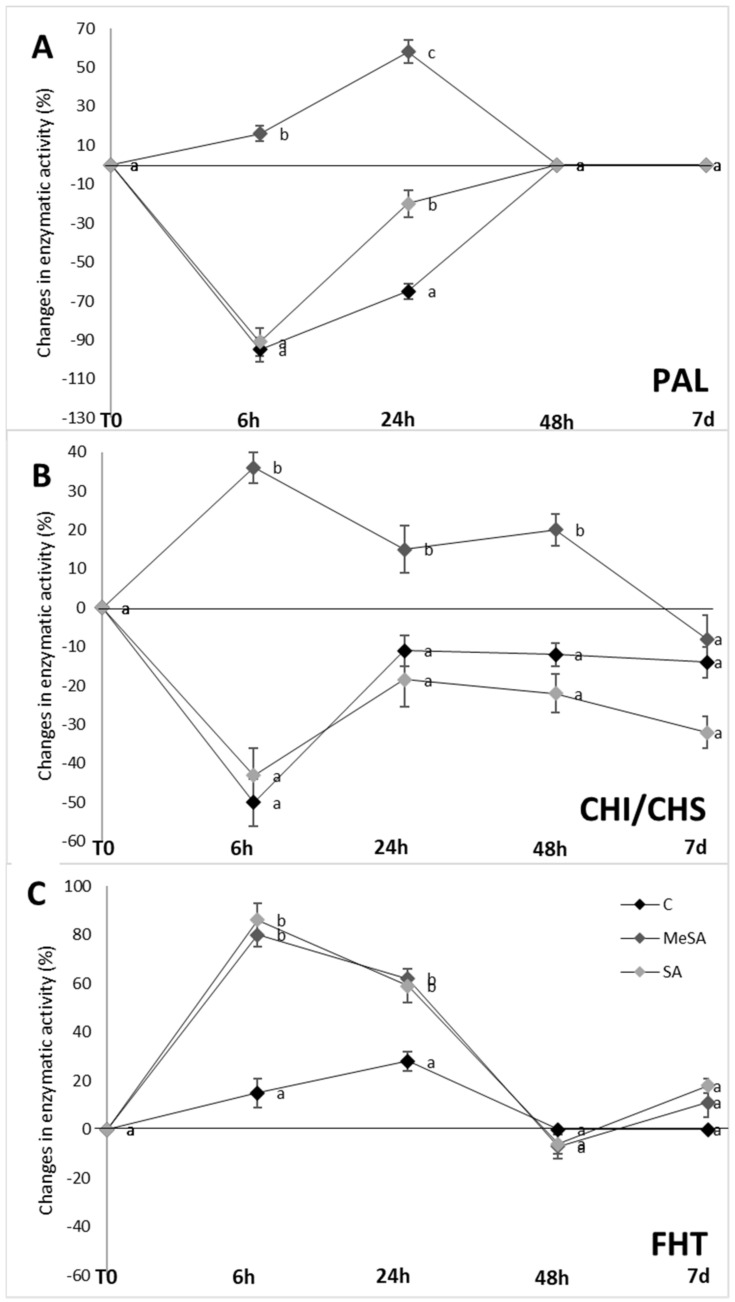
Specific enzyme activities in fruit peel for phenylalanine ammonia-lyase (**A**); (PAL), combined chalcone synthase (CHS) and chalcone isomerase (CHI) (**B**) and flavanone-3β-hydroxylase (**C**); (FHT) following control (**C**), methyl salicylic acid (MeSA) and salicylic acid (SA) treatments of apple trees, for the different sampling times after the first spraying (Spray#1). Data are means of calculated values ± standard error (*n* = 10). Different letters indicate significant difference between treatments at each sampling time (*p* < 0.05; Duncan tests).

**Table 1 plants-10-01807-t001:** Quantification of the major groups of phenolic compounds identified for the Spray#1 control and methyl salicylic acid (MeSA) and salicylic acid (SA) treatments for the apple fruit peel at the different sampling times. See Appendix A for quantification of the individual phenolics analyzed within these groups.

Phenolic Group	Treatment	Content (g/kg Dry Weight)				
		Before Spraying	After Spraying			
			6 h	24 h	48 h	7 days
Flavanols	Control	3.17 ± 0.03 ^a^	3.88 ± 0.19 ^a^	3.48 ± 0.17 ^a^	3.53 ± 0.40 ^a^	3.37 ± 0.22 ^a^
	MeSA	3.12 ± 0.01 ^a^	4.07 ± 0.58 ^a^	4.65 ± 0.50 ^b^	3.70 ± 0.31 ^a^	3.36 ± 0.15 ^a^
	SA	3.15 ± 0.02 ^a^	3.83 ± 0.24 ^a^	4.03 ± 0.12 ^ab^	3.56 ± 0.12 ^a^	3.22 ± 0.32 ^a^
Flavonols ^b^	Control	1.25 ± 0.08 ^a^	1.23 ± 0.07 ^a^	1.60 ± 0.22 ^a^	1.25 ± 0.04 ^a^	1.05 ± 0.02 ^a^
	MeSA	0.91 ± 0.07 ^a^	1.30 ± 0.12 ^a^	1.46 ± 0.17 ^a^	1.08 ± 0.12 ^a^	1.40 ± 0.07 ^b^
	SA	1.02 ± 0.13 ^a^	1.40 ± 0.11 ^a^	1.78 ± 0.10 ^a^	1.12 ± 0.10 ^a^	1.05 ± 0.02 ^a^
Hydroxycinnamic	Control	0.18 ± 0.01 ^a^	0.20 ± 0.00 ^a^	0.19 ± 0.02 ^a^	0.18 ± 0.01 ^a^	0.17 ± 0.01 ^a^
acids	MeSA	0.16 ± 0.01 ^a^	0.18 ±0.02 ^a^	0.23 ± 0.01 ^b^	0.18 ± 0.01 ^a^	0.22 ± 0.01 ^b^
	SA	0.16 ± 0.01 ^a^	0.19 ± 0.00 ^a^	0.17 ± 0.00 ^a^	0.17 ± 0.00 ^a^	0.17 ± 0.01 ^a^
Dihydrochalcones	Control	1.61 ± 0.08 ^a^	1.70 ± 0.05 ^a^	2.15 ± 0.20 ^a^	1.81 ± 0.02 ^a^	1.90 ± 0.05 ^a^
	MeSA	1.52 ± 0.16 ^a^	1.90 ± 0.19 ^a^	2.13 ± 0.17 ^a^	1.79 ± 0.08 ^a^	1.69 ± 0.10 ^a^
	SA	1.70 ± 0.14 ^a^	2.03 ± 0.02 ^a^	2.24 ± 0.08 ^a^	1.77 ± 0.12 ^a^	1.71 ± 0.02 ^a^
Anthocyanins	Control	0.34 ± 0.05 ^a^	0.43 ± 0.09 ^a^	0.55 ± 0.03 ^b^	0.44 ± 0.07 ^a^	0.90 ± 0.04 ^b^
	MeSA	0.26 ± 0.03 ^a^	0.27 ± 0.05 ^a^	0.31 ± 0.05 ^a^	0.24 ± 0.07 ^a^	0.39 ± 0.07 ^a^
	SA	0.28 ± 0.06 ^a^	0.47 ± 0.03 ^a^	0.46 ± 0.06 ^ab^	0.43 ± 0.07 ^a^	0.67 ± 0.08 ^b^
Total phenolics	Control	6.56 ± 0.22 ^a^	7.44 ±0.20 ^a^	8.64 ± 0.60 ^a^	7.21 ± 0.42 ^a^	7.55 ± 0.21 ^a^
	MeSA	5.97 ± 0.22 ^a^	7.73 ± 0.88 ^a^	7.81 ± 0.28 ^a^	6.98 ± 0.14 ^a^	6.84 ± 0.18 ^a^
	SA	6.30 ± 0.33 ^a^	7.91 ± 0.12 ^a^	7.67 ± 0.32 ^a^	7.04 ± 0.17 ^a^	6.82 ± 0.23 ^a^

Data are means ± standard error (*n* = 10). Different letters indicate significant differences among the treatments within each phenolic group and each sampling time (*p* < 0.05; Duncan tests).

## Data Availability

All data pertaining to this study is being held in computers owned by University of Ljubljana, Ljubljana, Slovenia, under control of the PI team.

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
