# Peer review of "Salicylic and Methyl Salicylic Acid Affect Quality and Phenolic Profile of Apple Fruits Three Weeks before the Harvest"

_plants, 2021, doi:10.3390/plants10091807_

Round 1

Reviewer 1 Report

I find the approach of this work is interesting in practical terms. You intend to test how the selected concentration of both SA and MeSA which applied 3 weeks before harvesting, can affect the apple fruit quality form different aspects like physical quality and some biochemical parameters

There are, though, some specific aspects of the experimental system which affect the interpretation of the results which I would like to point out:

In the title, it should be profile not profil, also I think it should be three weeks before the harvest (not last three week of maturation) as the complete maturation of apple fruit actually will be after harvest.

It was interesting to use one of SA or MeSA as one elicitor and using other type of elicitors because most of results did not show that much significance differences between SA and MeSA? What value to use both of them as preharvest? I suggest if possible to give a complete idea, you use them as pre and /or post-harvest?

Also, you had used specific concentrations 2 mM MeSA and 2 mM SA, 1 L/tree of both chemicals and it is usually known that these concentrations are more effect when it can be used postharvest by dipping fruit in it and when we spray usually the concentration should increase, there is no references in this regard (selected concentration) in the text

you used only 1 litter per tree?

What was the method to spry, it should be mentioned in the material and methods?

The physical quality was reduced by both of the chemical and also the class I fruit was reduced, and mostly of the references you were cited, showed the that the SA and/or MeSA can improve the physical quality like firmness, fruit weight, and fruit color. Can you explain how would these chemical can decrease the physical quality? However, it is reported that they can delay the ripping of the fruit which can keep the fruit quality higher for long time. Also, increasing the phenolic activity is good result but the value of this will be more if we find what could find the benefit behind this increase like, reducing chilling injuries during cold storage and infection. Because in this situation, as the appearance quality reduced, and increase in polyphenol, could be a sign of over ripping of the fruit, why we could say help fruit to overcome the stresses as we see firmness decreases?

You concluded that, SA and/or MeSA can improve the phenol profile of apple fruit, if you want to give a recommendation, which time is the best time to make this application? It is also need to explain why increase the phenol profile and it was coupled to decrease the physical quality, is there is any correlation analysis to find if there is relation?

Best regards

Author Response

Dear Reviewer 1!

Thank you for the review of this paper and the useful comments.

We have read the queries and we hope corrected the paper according to your suggestions. We have answered the comments, which we believe have helped us to improve the reports of our study for publication in Plants.

We look forward to your replay,

Authors.

POINT 1: In the title, it should be profile not profil, also I think it should be three weeks before the harvest (not last three week of maturation) as the complete maturation of apple fruit actually will be after harvest.

RESPONSE 1: Now that’s fixed. Thank you for your corrections. I marked the corrections in red in the text (row 2). New title: Salicylic and methyl salicylic acid affect quality and phenolic profile of apple fruits three weeks before the harvest.

POINT 2: It was interesting to use one of SA or MeSA as one elicitor and using other type of elicitors because most of results did not show that much significance differences between SA and MeSA? What value to use both of them as preharvest? I suggest if possible to give a complete idea, you use them as pre and /or post-harvest?

RESPONSE 2: While most articles have examined the effects of salicylic acid and other elicitors, we have focused on how various salicylic acid derivatives affect the quality of apples in the final stages of ripening and, in particular, the phenolic and enzymatic response. It seems to us that there are very few studies in the literature in which other salicylic acid derivatives have also been tested.

We were interested in whether quality (colour, firmness, fruit weight, yield, acid content, sugar content ...), phenolic response and enzymatic activity differed between apples treated with SA and between MeSA. We found that in our case, at least in MeSA, the phenolic content is higher, so only this could significantly contribute to the improvement of resistance to stressors with improved phenolic response. This was also confirmed by the improved activity of the enzyme PAL in MeSA treated apples, which is one of the key enzymes in phenolic synthesis.

MeSA definitely contributes to better phenolic content (with the exception of anthocyanins), so we believe that, at least in terms of better protection of apples with phytoalexins, spraying with MeSA before harvest is more successful. However, with respect to quality and postharvest treatments, we believe that further studies would be needed for a more robust explanation. I hope that with an additional explanation we have contributed to a better understanding of the whole idea.

POINT 3: Also, you had used specific concentrations 2 mM MeSA and 2 mM SA, 1 L/tree of both chemicals and it is usually known that these concentrations are more effect when it can be used postharvest by dipping fruit in it and when we spray usually the concentration should increase, there is no references in this regard (selected concentration) in the text

RESPONSE 3: The use of these two concentrations was conditioned by literature reviews and preliminary analyses. It is true that various articles also report higher concentrations for different fruits, especially for fruits with harder skin. We also performed a preliminary analysis where we found phytotoxic effects on the skin of apples sprayed 5 weeks before harvest. Concentrations of 1.5, 2.0, 3.0, 3.5, 4.0, and 4.5 mM were tested. As we found that a concentration of 3.0 (MeSA) can already cause burns on apples (5 weeks before harvest), we decided to use a concentration of 2 mM as we were particularly interested in the effects between SA and MeSA.

In addition, MeSA and SA were found to be poorly soluble in water at higher concentrations and the solutions caused burns to leaves. Of the concentrations tested, the 2 mM solution was selected because it dissolved well and did not cause phytotoxic reactions on leaves or fruits.

POINT 4: you used only 1 litter per tree?

RESPONSE 4: Yes.

POINT 5: What was the method to spry, it should be mentioned in the material and methods?

RESPONSE 5: Thank you for your warning, that this is missing. We added: with hand gasoline sprayer. We hope this is enough. (L306 in red).

POINT 6: The physical quality was reduced by both of the chemical and also the class I fruit was reduced, and mostly of the references you were cited, showed the that the SA and/or MeSA can improve the physical quality like firmness, fruit weight, and fruit color. Can you explain how would these chemical can decrease the physical quality? However, it is reported that they can delay the ripping of the fruit which can keep the fruit quality higher for long time. Also, increasing the phenolic activity is good result but the value of this will be more if we find what could find the benefit behind this increase like, reducing chilling injuries during cold storage and infection. Because in this situation, as the appearance quality reduced, and increase in polyphenol, could be a sign of over ripping of the fruit, why we could say help fruit to overcome the stresses as we see firmness decreases?

RESPONSE 6: Apple fruits were randomly selected during sampling, with no attention paid to visible external quality parameters. However, the treatments with MeSA resulted in lower fruit firmness. Whether this is related to faster ripening cannot be confirmed. Other analyzes conducted in the trial (sugar and organic acid content in the flesh) showed no significant differences between treatments. For a more accurate determination of the degree of ripeness, it would be necessary to determine the ethylene content in the fruit, which unfortunately there was no opportunity to do. According to the results obtained in the measurement of enzyme activity and phenolic content (in both cases MeSA was characterized by higher enzyme activity and phenolic content), it could be that a higher proportion of the primary metabolites was converted into phenolics, rather than to fruit grow.

POINT 7: You concluded that, SA and/or MeSA can improve the phenol profile of apple fruit, if you want to give a recommendation, which time is the best time to make this application? It is also need to explain why increase the phenol profile and it was coupled to decrease the physical quality, is there is any correlation analysis to find if there is relation?

RESPONSE 7: This was the first comprehensive study of its kind on the effects of salicylates on preharvest quality of apple trees. We conducted it only on the cultivar 'Topaz', a variety resistant to apple scab and rich in phenolic substances. In order to obtain more precise recommendations on when to treat and with what concentration of salicylates, additional studies should also be carried out on other apple cultivars. Based on our results, it appears that the greatest effect on phenolic content occurred 24 hours after treatment. How this affects other apple varieties would need further investigation. This recommendation could really improve our manuscript and might interest readers. We have added a few sentences to the conclusion. (Conclusion L449-L452 in red).

The quality of fruit can be degraded due to various environmental factors, which we can only try to eliminate as much as possible in our studies. However, in our case, it was the most surprisingly poorer coloration of the treated fruits due to the lower anthocyanin content. These (content of anthocyanins) did not improve with treatments like most of previous studies showed. Also, in some cases, this could happen when the plants are not in stressful situations. We have added the following reference (34) and paragraph to our article confirming this (L242-L252 in red):

In general, studies report anthocyanin content increased when treated with SA.11,12 However, there may also be a decrease in content under non-stressful conditions. This was shown by the results of a study by Bukhat et al,34 in which they found lower anthocyanin contents in plants treated with SA on non-saline soils, while the use of SA on stressed plants (saline soils) increased the anthocyanin content. They explained that stress factors increase the accumulation of ROS and that SA to detoxify ROS radicals would increase the production of anthocyanins as non-enzymatic antioxidants. When ROS does not need to be degraded, the level of stress-induced anthocyanins does not increase. The plants in our experiment were grown optimally, so there should be no stressful conditions. The fact that in our case anthocyanins did not increase in the treated apples can be attributed to this phenomenon.

I hope it will contribute to a better discussion.

Your comments were very valuable to the revision process. We have taken all of your comments into consideration and have changed our manuscript to better fit the framework of Plants.

Thank you for the time and effort that has been put into improving the quality of our work.

Reviewer 2 Report

The manuscript describes SA and MeSA spraying to apple trees during fruit maturation resulted in poor red coloration of fruit peel, smaller fruit size, and improved levels of some beneficial polyphenols (flavanols, flavonols and hydroxycinnamic acids) in fruit peel at 7 days after the first spraying #1(August 22). I have some concerns listed below.

  1. Previous studies have reported application of these salicylates resulted in increasing the anthocyanin and greater weights in apples or other fruit species. Why were the results in this study different from previous reports? Need to discuss it.
  2. How about the effects of the polyphenol contents and enzyme activities in final fruit products (harvest time)? Were there any effects on flesh tissue? Analysis of polyphenols were demonstrated only in fruit peel at before spraying, 6h, 24h 48h, and 7 days after the first spraying #1(August 22).

Small points

L97  The “maturation” includes several changes such as coloration, sugar contents, and acids. MeSA and SA accelerated the “decrease of firmness”.

Fig. 1  Indicate that T0 is before the first spray (Spraying #1) 

Fig. 3  How do authors detect “release” of the MeSA and SA contents?

         Were the same contents (µg/gDW) of SA and MeSA at spraying (0h) ?

         Indicate the X-axis label

L269-271  There was no data on PAL at the 24, 48, and 7 days after spraying in Figure 5.

Figure 5  Significance? Indicate the results of statistical analysis.

Author Response

Dear Reviewer 2!

Thank you for the review of this paper and the useful comments.

We have read the queries and we hope corrected the paper according to your suggestions. We have answered the comments, which we believe have helped us to improve the reports of our study for publication in Plants.

We look forward to your replay,

Authors.

POINT 1: Previous studies have reported application of these salicylates resulted in increasing the anthocyanin and greater weights in apples or other fruit species. Why were the results in this study different from previous reports? Need to discuss it.

RESPONSE 1: Thank you for your suggestion. Studies on the effect of preharvest treatment of fruit with salicylates on apple trees have not been made. We expected similar results to those reported by researchers in other fruit species. During sampling, we randomly selected fruit, but we did not pay attention to visible external quality parameters. With careful analyses of colour and anthocyanin content, we found that the application of MeSA in particular had a negative effect on both parameters, while the content of some other phenolic groups increased. This could be due to altered metabolism of the phenylpropanoid/flavonoid synthesis pathways. This is also suggested by the results of enzyme activities. To confirm this hypothesis definitively, future studies should include a larger number of enzymes and monitor their expression and activity.

I also added this paragraph in manuscript and I hope it will contribute to a better discussion (in red).:

In general, studies report anthocyanin content increased when treated with SA.11,12 However, there may also be a decrease in content under non-stressful conditions. This was shown by the results of a study by Bukhat et al,34 in which they found lower anthocyanin contents in plants treated with SA on non-saline soils, while the use of SA on stressed plants (saline soils) increased the anthocyanin content. They explained that stress factors increase the accumulation of ROS and that SA to detoxify ROS radicals would increase the production of anthocyanins as non-enzymatic antioxidants. When ROS does not need to be degraded, the level of stress-induced anthocyanins does not increase. The plants in our experiment were grown optimally, so there should be no stressful conditions. The fact that in our case anthocyanins did not increase in the treated apples can be attributed to this phenomenon.

PONIT 2: How about the effects of the polyphenol contents and enzyme activities in final fruit products (harvest time)? Were there any effects on flesh tissue? Analysis of polyphenols were demonstrated only in fruit peel at before spraying, 6h, 24h 48h, and 7 days after the first spraying #1(August 22).

RESPONSE 2: I thank you for your question. Hopefully, this answer will give you more insight into our experiment.

In this study, we analysed a phenolic profile in all collected samples (T0, 6h, 24h, 48h and 7 days after sprays 1, 2 and 3) in apple peel and also in apple flesh. Differences between treatments occurred (mostly) only after the first spraying, so in this manuscript we only consider the results after the first spraying. We also examined the enzymes only on these samples. We find that this is more interesting for the reader and also has more value in general. We also analysed the sugar and organic acid content of the apple, but again there were no significant differences.

POINT 3: L97 The “maturation” includes several changes such as coloration, sugar contents, and acids. MeSA and SA accelerated the “decrease of firmness”.

Thank you for the warning, we deleted this sentence.

POINT 4: Fig. 1 Indicate that T0 is before the first spray (Spraying #1)

It was corrected.

POINT 5: Fig. 3  How do authors detect “release” of the MeSA and SA contents?Were the same contents (µg/gDW) of SA and MeSA at spraying (0h). Indicate the X-axis label

It was corrected. Perhaps "release" is the wrong term here. It would be better if I used it: The contents in the apple peel? We determined this based on the measured concentration of SA and MeSA in the plant sample. Thank you for the warning.

POINT 6: L269-271 There was no data on PAL at the 24, 48, and 7 days after spraying in Figure 5.

Figure 5 was corrected. Now it is more clear.

POINT 7: Figure 5 Significance? Indicate the results of statistical analysis.

It was corrected.

Your comments were very valuable to the revision process. We have taken all of your comments into consideration and have changed our manuscript to better fit the framework of Plants.

Thank you for the time and effort that has been put into improving the quality of our work.

Reviewer 3 Report

The concentration of 2 mM MeSA and 2 mM SA were applied in this study. Please provide relevant evidence to support this concentration could be used in apple tree. It would be better to do different concentration treatment in apple tree.

In Figure 1C the data of SH side lack error bar. Please improve it.

In Figure 3 the data of SA lack error bar. Please improve it.

In Figure 4 the line of Y-axis was missing. Please improve it.

In Figure 5 all the date lack error bar. Please improve it.

Author Response

Dear Reviewer 3!

Thank you for the review of this paper and the useful comments.

We have read the queries and we hope corrected the paper according to your suggestions. We have answered the comments, which we believe have helped us to improve the reports of our study for publication in Plants.

We look forward to your replay,

Authors.

POINT 1: The concentration of 2 mM MeSA and 2 mM SA were applied in this study. Please provide relevant evidence to support this concentration could be used in apple tree. It would be better to do different concentration treatment in apple tree.

RESPONSE 1: Thank you fort his question. The use of these two concentrations was conditioned by literature reviews and preliminary analyzes. It is true that various articles also report higher concentrations for different fruits, especially for fruits with harder skin. We also performed a preliminary analysis where we found phytotoxic effects on the skin of apples sprayed 5 weeks before harvest. Concentrations of 1.5, 2.0, 3.0, 3.5, 4.0, and 4.5 mM were tested. As we found that a concentration of 3.0 (MeSA) can already cause burns on apples (5 weeks before harvest), we decided to use a concentration of 2 mM as we were particularly interested in the effects between SA and MeSA.

POINT2: In Figure 1C the data of SH side lack error bar. Please improve it.

RESPONSE 1:. It was corrected.

POINT 3: In Figure 3 the data of SA lack error bar. Please improve it.

RESPONSE 1:. It was corrected.

POINT4: In Figure 4 the line of Y-axis was missing. Please improve it.

RESPONSE 1: It was corrected.

POINT5: In Figure 5 all the date lack error bar. Please improve it.

RESPONSE 1:.It was corrected.

Your comments were very valuable to the revision process. We have taken all of your comments into consideration and have changed our manuscript to better fit the framework of Plants.

Thank you for the time and effort that has been put into improving the quality of our work.